# Kinetics of Serological Response in Patients with Severe Fever with Thrombocytopenia Syndrome

**DOI:** 10.3390/v13010006

**Published:** 2020-12-25

**Authors:** Sang Hyun Ra, Min Jae Kim, Min-Chul Kim, Se Yoon Park, Seong Yeon Park, Yong Pil Chong, Sang-Oh Lee, Sang-Ho Choi, Yang Soo Kim, Keun Hwa Lee, Sung-Han Kim, Sun-Ho Kee

**Affiliations:** 1Department of Infectious Diseases, Asan Medical Center, University of Ulsan College of Medicine, Songpa-gu, Seoul 05505, Korea; jesus4274@naver.com (S.H.R.); nahani99@gmail.com (M.J.K.); drchong@amc.seoul.kr (Y.P.C.); soleemd@amc.seoul.kr (S.-O.L.); sangho@amc.seoul.kr (S.-H.C.); yskim@amc.seoul.kr (Y.S.K.); 2Division of Infectious Diseases, Department of Internal Medicine, Chung-Ang University Hospital, Dongjak-gu, Seoul 06973, Korea; pour-soi@hanmail.net; 3Division of Infectious Diseases, Department of Internal Medicine, Soonchunhyang University College of Medicine, Yongsan-gu, Seoul 04401, Korea; sypark@schmc.ac.kr; 4Department of Infectious Diseases, Dongguk University Ilsan Hospital, Dongguk University College of Medicine, Ilsandong-gu, Goyang-si 10326, Korea; psy99ch@hanmail.net; 5Department of Microbiology, Hanyang University College of Medicine, Seongdong-gu, Seoul 04763, Korea; 6Department of Microbiology, Korea University College of Medicine, Seongbuk-gu, Seoul 02841, Korea; keesh@korea.ac.kr

**Keywords:** severe fever with thrombocytopenia syndrome (SFTS), viral load, immunoglobulin G (IgG), immunoglobulin M (IgM), immunofluorescence assay (IFA), enzyme-linked immunosorbent assay (ELISA)

## Abstract

Severe fever with thrombocytopenia syndrome (SFTS) is caused by SFTS virus (SFTSV). We investigated the detailed kinetics of serologic response in patients with SFTS. Twenty-eight patients aged ≥18 years were enrolled between July 2015 and October 2018. SFTS was confirmed by detecting SFTSV RNA in their plasma using reverse transcription polymerase chain reaction. SFTSV-specific IgG and IgM were measured using immunofluorescence assay (IFA) and enzyme-linked immunosorbent assay (ELISA). We found that SFTSV-specific IgG was detected at days 5–9 after symptom onset, and its titer was rising during the course of disease. SFTSV-specific IgM titer peaked at around week 2–3 from symptom onset. The SFTSV-specific seropositive rates for days 5–9, 10–14, 15–19, and 20–24 from symptom onset using IFA and ELISA were 63%, 76%, 90%, and 100%, and 58%, 86%, 100%, and 100%, respectively, for IgG, whereas they were 32%, 62%, 80%, and 100%, and 53%, 62%, 70%, and 100%, respectively, for IgM. The delayed IgM response could be attributed to the low sensitivity of SFTSV-specific IgM IFA or ELISA and/or impaired immune responses. The IgM test using IFA or ELISA that we used in this study is, therefore, insufficient for the early diagnosis of SFTS.

## 1. Introduction

Severe fever with thrombocytopenia syndrome (SFTS), which is caused by SFTS virus (renamed as Dabie bandavirus) in the genus *Bandavirus*, family *Phenuiviridae*, and order *Bunyavirales* [1], is an emerging endemic zoonosis usually transmitted by ticks, such as *Haemaphysalis longicornis* [2]. Since its first report in China in 2011 [2], SFTS was subsequently identified in South Korea and Japan in 2012, and reported in 2013 and 2014, respectively [3,4]. SFTS has become a significant threat to public health in East Asian countries with a high fatality rate of 16.2–32.6% [5,6], and effective antiviral therapy for SFTS virus (SFTSV) has not been available [7,8,9].

Previous studies [10,11] have reported that one of the major pathophysiological characteristics that causes high mortality in SFTS is related to cytokine storm. Additionally, other studies [12,13] revealed that the impairment of both innate and adaptive immune responses was associated with disease progression and mortality in SFTS. Song et al. [13] described a defect in serological responses to SFTSV that plays an important role in disease mortality rate, and impairment of both B and T cells could contribute to low anti-viral immunity. However, to the best of our knowledge, there is a huge paucity of clinical research data on the pattern of changes in antibody response throughout the course of this disease. We, therefore, investigated the detailed kinetics of anti-SFTSV antibody response in patients with SFTS.

## 2. Materials and Methods

### 2.1. Patients and Samples

Adult patients, aged ≥ 18 years, with confirmed SFTS, were enrolled at four university-affiliated hospitals in South Korea during the period of July 2015 to October 2018. The four hospitals were Asan Medical Center, Soonchunhyang University Seoul Hospital, Chung-Ang University Hospital, and Dongguk University Ilsan Hospital. SFTSV RNA in plasma specimens was confirmed using real-time reverse transcription polymerase chain reaction (RT-PCR). Patients’ plasma specimens in ethylenediaminetetraacetic acid (EDTA)-treated vacutainers were obtained during their hospitalization period. Plasma samples were immediately separated from whole blood and frozen at −80 °C until further analysis. The study protocol was approved by the Institutional Review Boards of each hospital (2016–127–0748 for Asan Medical Center, 2016–09–001 for Soonchunhyang University Seoul Hospital, 1970–002–376 for Chung-Ang University Hospital, and 2016–01–088 for Dongguk University Ilsan Hospital).

### 2.2. Quantification of SFTSV RNA

SFTSV RNA in plasma specimens was confirmed using RT-PCR as described in our previous study [14]. Briefly, RNA was extracted from the serum specimen using a viral RNA extraction kit (iNtRON Biotechnology, Gyeonggi, South Korea) according to the manufacturer’s instructions. To detect SFTSV RNA, the one-step RT-PCR was performed using a DiaStar 2× OneStep RT-PCR Pre-Mix kit (SolGent, Daejeon, South Korea) with the primers MF3 (5′-GATGAGATGGTCCAT GCTGATTCT-3′) and MR2 (5′-CTCATGGGGTGGAATGTCCTCAC-3′), and the following conditions: reverse transcription at 50 °C for 30 min and an initial denaturation step at 95 °C for 15 min, followed by 35 cycles each of denaturation at 95 °C for 20 s, annealing at 58 °C for 40 s, extension at 72 °C for 30 s, and a final extension step of 5 min at 72 °C.

### 2.3. Measurement of Anti-SFTSV IgG and IgM Using IFA

Serological tests for detecting anti-SFTSV IgG and IgM were performed using the immunofluorescence antibody assay (IFA), as previously described [15]. For IFA, Vero E6 cells infected with SFTSV were incubated in a 5% CO_2_ incubator at 37 °C for five days. Subsequently, cells were harvested, inoculated onto Teflon-coated well slides, and then fixed with acetone. IFA was carried out using a patient’s serum as the primary antibody and fluorescein-labeled antihuman IgG or IgM secondary antibodies (Thermo Fisher Scientific, Waltham, MA, USA). Serum specimens were diluted two-fold from 10 up to 2560. The incubation time of a patient’s serum for IgM detection was extended to 90 min, whereas 30 min incubation was performed for IgG detection. A monoclonal anti-SFTSV N antibody (manufactured in our laboratory) was used as the positive control. A titer of 80 was considered positive as applied in the previous study [16].

### 2.4. Measurement of Anti-SFTSV IgG and IgM Using ELISA

The enzyme-linked immunosorbent assays (ELISA) for anti-SFTSV IgM and IgG were performed as previously described [17]. To use the ELISA coating antigens, *E. coli*-expressed recombinant nucleoproteins (NP) were purified. A Nunc-Immuno Plate (Thermo Fisher Scientific, Waltham, MA, USA) was coated with a predetermined optimal quantity of NP (100 ng per well) and incubated overnight at 4 °C. An absorbance reader (Sunrise, Tecan’s Magellan™, Männedorf, Switzerland) was used to measure the absorbance in the ELISA. A sample was considered antibody-positive if it yielded an optical density (OD) above the predetermined cut-off value at 405 nm (OD_405_). The average OD values for the negative controls were between 0.18 and 0.24. The IgM and IgG antibody-positive serum of an SFTS patient, presenting positive results in the immunofluorescence assay (IFA), was used as a positive control (OD_405_ of IgM = 1.36–1.95 and OD_405_ of IgG = 1.36–1.76). The anti-SFTSV IgM and IgG levels in the specimens obtained from the patients were considered positive if they were higher than the positive control (OD_405_ = 1.36).

## 3. Results

### 3.1. Clinical Characteristics of the Patients

A total of 28 patients with confirmed SFTS were enrolled in this study. IFA for both IgG and IgM were performed on all patients during the course of the disease. Among these patients, 14 (50%) were men, and their mean age (±standard deviation) was 61.8 (±9.9) years. Detailed clinical characteristics of the patients in this study are shown in Table 1.

In addition, three patients (10.7%) died due to aggravated multi-organ failure during the course of this disease. Detailed clinical characteristics and serologic response of these three deceased patients are shown in Appendix A, respectively. Compared to survivors (*n* = 25), only viral load at days 5–9 from symptom onset was statistically high in the non-survivors group (*p* = 0.008). Other parameters such as age, underlying diseases, and antibody response did not show significant differences between survivors and non-survivors groups.

### 3.2. Kinetics of Viremia and Antibody Responses in Patients with SFTS

Viral load peaked on days 5–9 after symptom onset, and then gradually decreased over the course of disease. Positive IgG titers against SFTSV using both IFA and ELISA were detected on days 5–9 after symptom onset, with a robust antibody response (≥1280) for IFA detected around week 3 after symptom onset. Anti-SFTSV IgM for IFA reached its peak level around week 3 after symptom onset. Although, for ELISA, anti-SFTSV IgG and IgM patterns based on disease course were similar, the IgM peak occurred 5 days earlier than for IFA. The detailed kinetic data of viral load and antibody responses are shown in Figure 1 (curved plot) and Figure 2 (scattered plot).

The median times from symptom onset to the last detection of positive viremia to negative viremia, when patients were negative for viral load at discharge, were 14 days (IQR: 10.05–22.50) and 16 days (IQR: 13.00–23.75), respectively. In addition, the median time from the symptom onset to the positive IgG response and the robust IgG antibody response using IFA was 9 days (IQR: 6.00–11.25) and 17 days (IQR: 13.25–20.75), respectively. The median time from the symptom onset to the positive IgG response using ELISA was 9 days (IQR: 6.50–11.00).

### 3.3. Seropositive Rate of IgG and IgM Based on Time Course of SFTS Disease

The SFTSV-specific IgG seropositive rates at cut-off values of 80 for IFA and OD_405_ = 1.36 for ELISA for day 5–9, 10–14, 15–19, and 20–24 after symptom onset were: 63%, 76%, 90%, and 100%, and 58%, 86%, 100%, and 100%, respectively. The SFTSV-specific IgM positive rates for IFA and ELISA for days 5–9, 10–14, 15–19, and 20–24 after symptom onset were: 32%, 62%, 80%, and 100%, and 53%, 62%, 70%, and 100%, respectively (Table 2). One (3.6%) out of the 28 patients showed positive IgG response (titer cut-off value at 80 for IFA, and OD_405_ = 1.36 for ELISA) but was negative for IgM using both IFA and ELISA at day 9 from symptom onset. The detailed data depending on sampling times are shown in Appendix A. In the subgroup that excluded the patients (*n* = 14) in whom only early blood samples within 2 weeks from the symptom onset were available, the sensitivities of IgG and IgM IFA were 100% and 86%, respectively, and the sensitivities of IgG and IgM ELISA were 100% and 93%, respectively.

## 4. Discussion

Since the detection of SFTSV in 2009 in China, SFTS has become a public health threat in East Asian countries. One of the biggest concerns related to SFTS is the high mortality rate due to the lack of effective treatment. Even up to this point, pathogenesis and immune responses of SFTS are still poorly understood.

A prior study [18] suggested that anti-SFTSV IgM levels could be detected at a medium of 9 days, peaking at week 4, and persisted until six months after disease onset; anti-SFTSV IgG could still be detected at a medium of six weeks and peaked at six months. Another study [13] reported that SFTSV NP-specific IgM was detected at an early phase after symptom onset, persisting throughout hospitalization in all recovered patients, with SFTSV NP-specific IgG detected from week 2 after symptom onset. However, our data showed that SFTSV-specific IgM using IFA was detected only at days 15–19 after symptom onset, and peaked at around week 3. However, SFTSV-specific IgM using ELISA was detected at days 5–9 from symptom onset, and peaked at days 15–19. Additionally, SFTSV-specific IgG was detected at days 5–9 after symptom onset, with a robust antibody response (≥1280) using IFA observed at around week 3. Moreover, one of the 28 patients was positive for IgG, but not for IgM, at day 9 from symptom onset. These results indicate that IgM serological tests using IFA or ELISA alone is inadequate for the early diagnosis of SFTS.

Taken together, these results indicate that careful consideration should be given to the issue of delayed IgM response to SFTSV in the course of this disease. It is well known that activated naïve B cells, after infection or immunization, can polarize their surface IgM and IgD to express IgG, IgA or IgE through immunoglobulin class switching [19]. Therefore, although it is possible that delayed IgM response to SFTSV is associated with the sensitivity of SFTSV-specific IgM IFA, we could not rule out the possibility of SFTSV-induced alterations in host immune response. In particular, one of the aforementioned studies [13] suggested that interleukin (IL)-10 contributes to the impaired B cell immunity by inhibiting germinal center formation and differentiation of dendritic cells. This finding indicates that impaired B cells may affect the immunoreactivity of IgM to SFTS virus. Therefore, in contrast to other studies, our data showed a delayed IgM response compared to that of IgG. However, this finding may be associated with the following attributes of the cohort: a small sample size, experimental challenges due to the sensitivity of SFTSV-specific IgM IFA or ELISA, and/or attenuated adaptive immune response to SFTS. Thus, further in-depth studies with more sensitive serological tests for SFTSV will answer this important phenomenon.

Nevertheless, there were several limitations in our present study. First, most of the sera specimens were collected on 5–19 days from symptom onset (Appendix A). For this reason, it was difficult to evaluate the antibody response in the recovery phase of SFTS. Therefore, further well-designed studies with long-term follow-up for SFTS are required. Second, we had a relatively small number of patients (*n* = 28), including only three deceased patients with different specimen collection dates. For these reasons, there seems to be no significant difference between the survivor and non-survivor groups. Third, we did not perform any experiment on cell-mediated immune response which could have provided some important highlights on the mechanism of SFTSV infection. Fourth, we determined the cut-off value of OD in ELISA using positive controls rather than standard methods such as the average of negative controls plus three times the standard deviation (mean + 3SD) or receiver operator characteristic (ROC) analysis. Thus, the relatively high cut-off value of the ELISA we used might result in the underestimation of the sensitivity of this assay. However, this did not substantially alter our main findings in terms of the results of suboptimal sensitivity of SFTSV-specific IgM IFA during the early course of SFTS patients. Finally, we did not demonstrate that the IFA or ELISA we used were sensitive enough to detect small amount of SFTSV-specific antibodies. Therefore, it is difficult to draw a firm conclusion that the IgM test using IFA or ELISA is insufficient for the early diagnosis of SFTS. Further studies based on more robust designs, focusing on cellular and humoral immune responses against SFTSV, will be of pivotal importance in the quest for developing vaccines and antiviral agents.

In conclusion, our data suggest that SFTSV-specific IgM response was relatively delayed compared to that reported in previous studies. This may be related to the low sensitivity of SFTSV-specific IgM IFA or ELISA, and/or impaired adaptive immune response. Thus, the SFTSV-specific IgM IFA or ELISA that we used in this study are insufficient for the early diagnosis of SFTS, and further robust studies are required to elucidate the humoral and cellular immune responses against SFTSV.

## Figures and Tables

**Figure 1 viruses-13-00006-f001:**
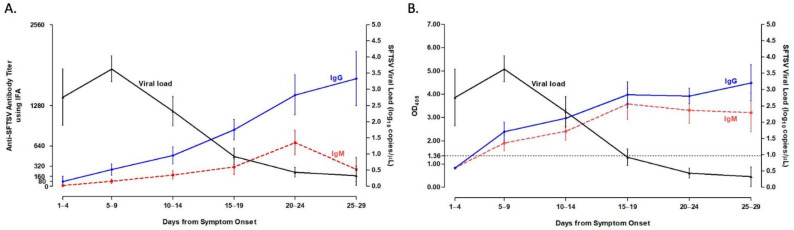
Kinetics of viremia and antibody responses in patients with SFTS. Data are denoted as means with standard error of mean (SEM). Both IgG and IgM were measured using IFA (**A**) and ELISA (**B**).

**Figure 2 viruses-13-00006-f002:**
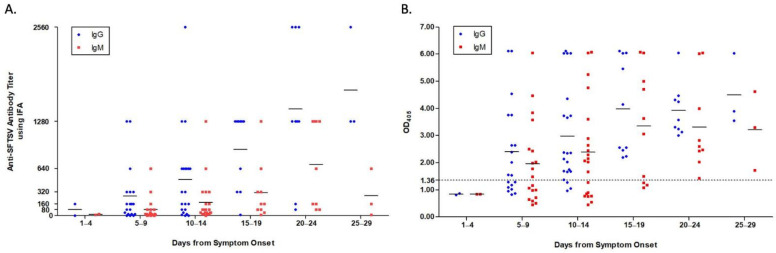
Scattered plot of antibody responses in patients with SFTS. Data are denoted as means. Both IgG and IgM were measured using IFA (**A**) and ELISA (**B**).

**Table 1 viruses-13-00006-t001:** Clinical characteristics of the studied patients.

**Characteristics**	**No. of Patients (*n* = 28) ***
**Age (years), mean ± SD**	61.8 ± 9.9
**Sex**	
Male	14 (50)
Female	14 (50)
**Season (months)**	
Spring-Summer (3–8)	14 (50)
Fall (9–11)	14 (50)
**Clinical characteristics**	
Fever	28 (100)
Tick- or chigger-bite wound	8 (28.6)
Skin rash	4 (14.3)
Bleeding	5 (17.9)
Myalgia	15 (53.6)
Anorexia/General weakness	21 (75)
Nausea/Vomiting	14 (50)
Abdominal pain	6 (21.4)
Diarrhea	14 (50)
Cough/Sputum/Dyspnea	9 (32.1)
Altered mental status	13 (46.4)
**Underlying diseases**	
Previously healthy	12 (42.9)
Diabetes mellitus	9 (32.1)
Solid tumor	2 (7.1)
Hematologic malignancy	0
Chronic liver disease	1 (3.6)
Chronic kidney disease	0
Chronic lung disease	3 (10.7)
Autoimmune disease	1 (3.6)
Solid organ transplantation	0
Hematopoietic stem cell transplantation	0
Immunosuppressant	0
**Clinical courses**	
General ward admission	18 (64.3)
ICU admission	10 (35.7)
In-hospital mortality	3 (10.7)
**Treatments**	
Doxycycline	23 (82.1)
Ribavirin	14 (50)
Plasma exchange	19 (67.9)
Convalescent plasma therapy	2 (7.1)
Self-limiting	1 (3.6)
	**Median (IQR)**
**Laboratory findings**	
White blood cell (/μL)	1800 (1050–2748)
Neutrophil (%)	56 (46–69)
Lymphocyte (%)	35 (25–48)
Monocyte (%)	6 (3–10)
Hemoglobin (g/dL)	14.0 (12.4–15.0)
Platelet (×10^3^/μL)	60 (42–78)
BUN (mg/dL)	15.0 (10.5–23.8)
Creatinine (mg/dL)	0.90 (0.62–1.13)
AST (IU/L)	204 (131–429)
ALT (IU/L)	95 (69–199)
CRP (mg/dL)	0.4 (0.1–0.9)
PT INR ^†^	1.05 (0.99–1.11)
aPTT (seconds) ^†^	39.9 (37.2–46.3)

Abbreviations: ALT, alanine aminotransferase; aPTT, activated partial thromboplastin time; AST, aspartate aminotransferase; CRP, C-reactive protein; ICU, intensive care unit; INR, international normalized ratio; IQR, interquartile range; PT, prothrombin time; SD, standard deviation. * All clinical characteristics and laboratory data were obtained when the patients first visited the emergency room or outpatient clinic. Data represent number of patients (%) unless otherwise specified. ^†^ PT INR and aPTT were unavailable in three patients.

**Table 2 viruses-13-00006-t002:** Seropositive rates of SFTSV-IgG and -IgM based on patients’ clinical course of disease.

**For IgG** *						
**Days from Symptom Onset**	**1–4**	**5–9**	**10–14**	**15–19**	**20–24**	**25–29**
IFA (≥80)	1/2 (50%)	12/19 (63%)	16/21 (76%)	9/10 (90%)	9/9 (100%)	3/3 (100%)
ELISA (OD_405_ ≥ 1.36)	0/2 (0)	11/19 (58%)	18/21 (86%)	10/10 (100%)	9/9 (100%)	3/3 (100%)
**For IgM** *						
**Days from Symptom Onset**	**1–4**	**5–9**	**10–14**	**15–19**	**20–24**	**25–29**
IFA (≥80)	0/2 (0)	6/19 (32%)	13/21 (62%)	8/10 (80%)	9/9 (100%)	2/3 (67%)
ELISA (OD_405_ ≥ 1.36)	0/2 (0)	10/19 (53%)	13/21 (62%)	7/10 (70%)	9/9 (100%)	3/3 (100%)

Abbreviations: ELISA, enzyme-linked immunosorbent assay; IFA, immunofluorescence assay; IgG, immunoglobulin G; IgM, immunoglobulin M; OD, optical density. * Numerator and denominator represent a number of seropositivity and a number of available data, respectively.

## Data Availability

The data presented in this study are available in this article and Appendix A.

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
