# Peer review of "Kinetics of Serological Response in Patients with Severe Fever with Thrombocytopenia Syndrome"

_viruses, 2020, doi:10.3390/v13010006_

Round 1
Reviewer 1 Report
This is a study to analyze the antibody responses of IgG and IgM in patients with SFTS by IFA and ELISA in Korea. The authors described that IgM response in the subjects occurred late after the onsets of disease and detection of IgM may not be useful for the diagnosis of SFTS.
Major points
- The sensitivities of IFA and ELISA used in this study are not described in this manuscript. As authors described in the section of Discussion, whether antibody is detectable or not in a patient at certain time point depends on them. Indeed, antibody to SFTSV was reported to be positive several days after the onsets in patients with SFTS by a sensitive method such as double-antigen ELISA (Umeki eta l. JVM, 2020). Therefore, based on the data shown in this manuscript, it is difficult to conclude that detection of antibodies to SFTSV is not sufficient for the early diagnosis of infection.
- Number of subjects were described as 28; however, only 19 or 21 of 28 were tested even during days 5-19. The numbers of test at the other time points were equal or less than ten. These numbers are too small to show the true positive rate of antibody at each time point.
Minor point
Reference 16 and 17 seem to be a same paper.
Reviewer 2 Report
Overall, the manuscript reads well and appears to be scientifically sound. However, a few grammatical errors should be addressed (written on attached file), as well as more information for the patients that died.
Specific comments:
- Line 46: An SFTS patient that died in the hospital in 2012 was diagnosed retrospectively in 2013. Therefore, the first case identified in Korea was in 2012 and reported in 2013.
- Table 1: The heading indicates No. (%). However, the numbers in ( ) under laboratory findings are not %, but a range. Perhaps make a separate heading for this and place this at the bottom of the Table, rather than in the middle.
- Table 1: Under Clinical course there is only 13 patients identified. List the others as outpatients (?) or hospitalized (?),
- The 3 patients that died, there is no information. List the information as to the time of death after the onset of symptoms and IgM and IgG levels at the time of death. Did their IgM/IgG levels progress at a different rate than patients that survived? What was their age? Since mortality rates are higher in older populations. Were there underlying causes, e.g., diabetes? I understand that this paper is concerned with the detection of SFTSV using IFA and ELISA, but there may be differences associated with these for patients that die due to the disease. Albeit, the sample size for the patients that died is very small.
- Another limitation, as defined, is the small sample size. From the dot diagram, the number of patients followed to days 25+ was only 3. However, the significance is when 100% of the patients were positive by both methods. This should be addressed.
- References: Minor errors identified on the manuscript.

Round 2
Reviewer 1 Report
Authors showed the detection rates of antibodies in the subjects in the table of revised manuscript, which are not the sensitivity of the tests they used. Unless the authors can show that the tests they used in this study are sensitive enough to detect small amount of antibody, it is not adequate to conclude that IgM test using IFA or ELISA alone is insufficient for the early diagnosis of SFTS in general. Based on the data shown in this manuscript, it is better to say that early diagnosis of SFTS was not possible only by the detection of IgM antibody using methods describe in this manuscript.
